# Viscosupplementation and Synovial Fluid Rheology: A Hidden Risk for Bacterial Biofilm Formation in Joint Infections?

**DOI:** 10.3390/microorganisms13040700

**Published:** 2025-03-21

**Authors:** Fabiana Giarritiello, Luigi Regenburgh De La Motte, Lorenzo Drago

**Affiliations:** 1UOC Laboratory of Clinical Medicine with Specialized Areas, IRCCS MultiMedica, 20138 Milan, Italy; f.giarritiello@studenti.unimol.it (F.G.); luigi.delamotte@multimedica.it (L.R.D.L.M.); 2Department of Medicine and Health Sciences “V. Tiberio”, University of Molise, 86100 Campobasso, Italy; 3Department of Biomedical Health Sciences, University of Milan, 20133 Milan, Italy

**Keywords:** synovial fluid, rheology, viscosupplementation, hyaluronic acid, septic arthritis, periprosthetic joint infection, vascular impairment, bacterial biofilm

## Abstract

Synovial fluid (SF) plays a critical role in joint lubrication, load distribution, and maintaining homeostasis within the synovial cavity. Its rheological properties, primarily influenced by hyaluronic acid (HA) and other macromolecules, are essential for normal joint function. However, alterations in the physicochemical characteristics of SF can occur due to septic conditions, including septic arthritis (SA) and periprosthetic joint infections (PJIs), which significantly impact joint health. Bacterial colonization in infected joints often leads to the formation of biofilms, microbial aggregates encased in an extracellular matrix, which confer resistance to antibiotics and host immune responses. Biofilm formation in SF-altered environments is a major challenge in treating joint infections, particularly in patients with prosthetic implants. *Viscosupplementation*, primarily through intra-articular hyaluronic acid (HA) injections, has been widely used to restore SF viscosity and function in degenerative joint diseases. More recently, polyacrylamide (PAA)-based gels have emerged as an alternative viscosupplementation strategy. However, concerns have been raised regarding the potential impact of viscosupplements on biofilm formation and bacterial adhesion in septic joints, as changes in SF viscosity and composition may influence bacterial colonization and persistence. This review aims to assess the interaction between viscosupplementation and biofilm formation in septic joint pathologies, examining the effects of HA and PAA on SF rheology and bacterial adhesion. Understanding these interactions is crucial for optimizing therapeutic strategies and mitigating the risk of biofilm-associated infections in patients undergoing viscosupplementation.

## 1. Introduction

Synovial fluid (SF) plays a fundamental role in joint health by acting as a lubricant, reducing friction and distributing mechanical loads within the synovial cavity [1]. The viscoelastic behavior of SF is primarily determined by the concentration and molecular weight of hyaluronic acid (HA), which allows SF to exhibit non-Newtonian properties, adapting dynamically to different mechanical loads [2]. This behavior is essential for joint function as it enables SF to reduce shear stress while maintaining a protective environment for cartilage surfaces [3]. Articular cartilage, being avascular and aneural, relies on SF not only for mechanical protection but also for nutrient exchange and metabolic homeostasis [4]. The surface of cartilage is coated by a specialized lubricating layer composed of HA, proteoglycan 4 (PRG4, also known as lubricin), and phospholipids, which act synergistically to minimize shear forces and maintain joint function [2]. Alterations in the rheological properties of SF, such as decreased HA concentration or enzymatic degradation, can compromise this protective mechanism, leading to increased cartilage wear and altered biomechanical stress distribution [3].

In certain clinical conditions, such as septic arthritis (SA) and post-surgical states, SF undergoes notable rheological changes due to the presence of pathogenic bacteria [5,6]. These changes are often exacerbated by therapeutic interventions like viscosupplementation, a widely used treatment to restore SF viscosity in degenerative joint diseases such as osteoarthritis [7]. However, modifying the biomechanical properties of SF in infection-prone joints may have unintended effects. The progressive increase in SF density due to viscosupplements could create a high-viscosity environment that acts as a quasi-surface, inadvertently promoting *biofilm* formation even in its planktonic state [8]. This is of particular concern since biofilms—structured bacterial communities encased in a self-produced extracellular matrix—are notoriously resistant to antibiotics and immune responses, making infection management significantly more challenging [9].

Viscosupplementation, primarily through intra-articular HA injections, has been extensively used to alleviate joint degeneration, while polyacrylamide (PAA)-based gels have recently emerged as an alternative strategy [10]. While these approaches aim to improve joint function and provide mechanical protection, their potential role in facilitating bacterial adhesion and biofilm formation remains unclear. In addition to its rheological and lubricating properties, HA has been increasingly recognized for its senomorphic effects, modulating chondrocyte metabolism and reducing cellular senescence in osteoarthritic joints [11]. The relationship between SF rheology, viscosupplementation, and biofilm development has not been fully elucidated, raising concerns about the long-term implications of these therapies in infection-prone joints. Given these uncertainties, this review aims to explore the interplay between SF rheology and bacterial colonization, with a particular focus on how viscosupplements may influence infection susceptibility and biofilm formation. By critically analyzing the impact of viscosupplementation on SF properties, we seek to assess the associated risks and provide insights into optimizing therapeutic approaches to mitigate infection-related complications.

## 2. Materials and Methods

To select the most relevant articles for this narrative review, a structured search strategy was implemented using Google Scholar and PubMed, with predefined inclusion and exclusion criteria. While PRISMA flowcharts are typically used for systematic reviews, we have adapted and included a PRISMA-inspired flow diagram to enhance transparency, summarizing the article identification and selection process (Figure 1).

To select the most relevant articles for this narrative review, an extensive literature search was conducted using the Google Scholar and PubMed databases, selecting studies published in the last 10 years, in English, and from peer-reviewed journals. The research was divided into three phases to comprehensively explore the topic of infection risk and biofilm formation following the use of viscosupplements.

The first phase focused on joint infections and biofilm formation, using the following search query, (“infectious arthritis” OR “septic arthritis” OR “periprosthetic joint infection” OR (“rheumatoid arthritis” AND “secondary infection”)) AND (“biofilm” OR “bacterial adhesion”) AND “synovial fluid”, which identified 46 studies in Google Scholar and 14 in PubMed.

The second phase investigated the effects of viscosupplements on synovial fluid rheology and infections, using the query, (“hyaluronic acid” OR “polyacrylamide”) AND (“synovial fluid rheology” OR “synovial fluid viscosity”) AND (“bacterial biofilm” OR “bacterial colonization” OR “joint infection”), which retrieved 28 studies in Google Scholar, while no relevant studies were found in PubMed.

Finally, the third phase analyzed the role of synovial fluid rheology in bacterial growth modulation, using the query, (“synovial fluid rheology” OR “synovial fluid viscosity”) AND (“bacterial infection” OR “septic arthritis” OR “biofilm formation”) AND (“hyaluronic acid” OR “viscosupplementation” OR “extracellular matrix”), which obtained 28 studies in Google Scholar and 1 in PubMed.

Articles were selected based on strict criteria, with only original scientific studies with experimental or clinical data included, while literature reviews, duplicate studies, animal model research, opinion articles, and letters to the editor, were excluded. The selection process was conducted in two stages:Initial screening based on title and abstract to identify relevant studies.Full-text analysis to exclude articles lacking clinically relevant or methodologically robust data.

After completing the selection process, 11 articles were included and analyzed in this review.

## 3. Synovial Fluid Rheology and Infection Susceptibility

### 3.1. Healthy Synovial Fluid Rheology: Viscosity and Elasticity

Synovial fluid (SF) exhibits unique rheological properties that are essential for maintaining joint function. As a viscoelastic biofluid, it provides both lubrication and mechanical protection, ensuring smooth articulation and reducing cartilage wear. The key components that regulate SF’s rheological behavior are hyaluronic acid (HA), lubricin, and plasma-derived proteins such as albumin and fibrinogen [4,8,12,13]. SF is a non-Newtonian fluid, and this property allows SF to remain highly viscous at rest, providing cartilage protection, while reducing viscosity during movement, thereby facilitating joint articulation [4,14]. The molecular weight of HA is a key determinant of these rheological characteristics, as high-molecular-weight HA (~$10^{7}g/mol)$ forms an entangled molecular network that enhances shock absorption and lubrication [15]. This relationship can be quantitatively described by the power law equation for apparent viscosity, which is as follows:$$\eta_{app}=K{\dot{\gamma }}^{n-1}$$
where $\eta_{app}$ is the apparent viscosity [Pa·s], K it is the consistency coefficient [$Pa\cdot s^{n}$], $\dot{\gamma }$ is the rate of deformation (shear rate) [$s^{-1}$], and n is the index of fluid behavior (dimensionless). For a shear-thinning fluid, 0<n<1, meaning that as shear rate increases, viscosity decreases. This relationship has been mathematically described, highlighting how synovial fluid viscosity adapts dynamically under different velocity gradients, transitioning from shear-thinning to more Newtonian-like behavior over time in specific pathological conditions [3]. Additionally, SF contains proteins at concentrations of 10–30 mg/mL, and its elastic modulus G’ is stabilized between $10^{0}$ and $10^{3}$ $s^{-1}$, with a relaxation time constant (τ) of approximately ≈50–100s [1] The interaction between HA and lubricin (PRG4) further optimizes lubrication efficiency, minimizing boundary friction and supporting joint homeostasis. These rheological properties play a crucial role in nutrient exchange, waste removal, and joint stability. The synergistic relationship between SF and cartilage ensures an optimal low-friction environment, reducing mechanical wear and preserving articular integrity [16]. Any alteration in HA concentration, molecular weight, or PRG4 distribution can lead to significant rheological changes, predisposing joints to mechanical stress, potential inflammation, and consequent predisposition to infection [4].

### 3.2. Rheology of Pathological Synovial Fluid in Infected Joints

Pathological conditions such as septic arthritis (SA) and periprosthetic joint infections (PJIs) induce significant alterations in synovial fluid (SF) composition and rheology due to bacterial colonization and the resulting inflammatory responses [5,6,17]. These changes impair SF’s natural lubrication and protective functions, exposing the joint to increased mechanical stress and cartilage degradation. Infected SF exhibits a substantial increase in leukocyte count and inflammatory cytokines (IL-6, TNF-α, IL-1β), which trigger a cascade of immune responses that exacerbate tissue damage [18,19]. A primary consequence of infection is the degradation of hyaluronic acid (HA), which leads to the following:

A reduction in HA concentration and molecular weight, shifting from $10^{7}g/mol$(in healthy SF) to values as low as $10^{6}\;g/mol$, resulting in a drastic loss of viscoelastic properties.An increased protein content (albumin, fibrinogen) that elevates friction and contributes to cartilage wear.An altered pH and osmolarity, creating a microenvironment conducive to bacterial proliferation and biofilm formation [12,20].

Under normal conditions, SF exhibits shear-thinning behavior, meaning its viscosity decreases with increasing shear rate. However, in infected SF, HA degradation reduces zero-shear viscosity ($\eta_{0}$) from 1–40 Pa·s (in healthy SF) to as low as 0.004–1 Pa·s. This weakens the shear-thinning effect, shifting the fluid toward a more Newtonian behavior (increase in n closer to 1), making the fluid behave more like a Newtonian liquid. Concurrently, the elastic modulus (G’) loses its plateau, indicating a loss of elasticity. At the same time, the relaxation time (τ) decreases to 1–20 s, reflecting structural instability within the entangled HA network [1]. These progressive rheological changes are consistent with mathematical modeling approaches that predict a shift from non-Newtonian to Newtonian behavior as viscosity drops in response to inflammatory degradation and enzymatic activity [3]. This process occurs through multiple mechanisms, as Figure 2 graphically shows. Oxidative stress, particularly involving metal ions such as copper and iron, generates free radicals that contribute to HA fragmentation, altering its molecular integrity. Additionally, enzymatic depolymerization by bacterial and host enzymes breaks HA into smaller fragments, diminishing its ability to maintain the fluid’s viscoelastic properties. Structural changes further exacerbate this degradation, as HA chains transition from an entangled network to a more compact, less functional structure, reducing viscosity and impairing shock absorption. Consequently, SF loses its capacity to evenly distribute mechanical loads across the joint, increasing cartilage surface stress and accelerating joint degeneration [4,12].

## 4. Managing HA Imbalance: Viscosupplement, a Therapeutic Challenge

The necessity of viscosupplementation arises from the direct correlation between SF deterioration and the depletion of HA content. In patients with osteoarthritis or joint trauma, HA concentration can be reduced by up to 50% compared to healthy individuals, and its average molecular weight can also decline significantly [12,20]. These experimental observations highlight the need for viscosupplementation solutions aimed at restoring the rheological behavior of SF, ensuring that the crossover values, elastic modulus, and viscous modulus are approximate to those found in the synovial fluid of a healthy individual.

However, while viscosupplementation effectively restores lubrication and mechanical protection, excessive HA administration poses potential risks. A delicate balance is required to ensure that SF maintains its optimal rheological properties without fostering bacterial adhesion and biofilm formation. Understanding the interplay between SF composition, viscosity, and microbial dynamics is essential for refining viscosupplementation protocols and minimizing infection risks in joint diseases and prosthetic interventions [4,12].

### 4.1. Hyaluronic Acid Injections: History, Mechanism, and Clinical Use

The concept of viscosupplementation was pioneered in the mid-20th century following extensive research on synovial fluid rheology. One of the earliest proponents, the Hungarian scientist Endre A. Balazs, hypothesized that intra-articular injections of viscoelastic solutions could restore the physiological properties of synovial fluid in diseased joints. His findings laid the foundation for viscosupplementation as a therapeutic intervention aimed at restoring joint lubrication, reducing pain, and improving overall joint function [8,12].

Since then, viscosupplementation has gained widespread clinical acceptance, particularly in managing osteoarthritis (OA) [5]. HA injections have also been explored in post-traumatic joint conditions, inflammatory arthritis, and post-surgical recovery protocols [4,20]. Hyaluronic acid (HA) is an essential glycosaminoglycan naturally present in synovial fluid, where it provides lubrication, viscoelasticity, and chondroprotection. It plays a crucial role in shock absorption and the distribution of mechanical loads across the joint surface [15]. In degenerative conditions, HA concentration and molecular weight are significantly reduced, leading to a decline in synovial fluid viscosity, increased cartilage wear, and heightened susceptibility to inflammation and pain [12].

HA injections aim to restore these biomechanical properties by increasing synovial fluid viscosity, thereby improving lubrication and reducing friction during joint movement [21]. Beyond its mechanical function, HA exhibits anti-inflammatory and chondroprotective properties, acting through multiple pathways, which are as follows:Interaction with CD44 and RHAMM receptors: HA binds to these receptors on chondrocytes and synovial cells, modulating inflammation and promoting extracellular matrix production while influencing cellular senescence [4,22].Cytokine inhibition: HA has been shown to inhibit pro-inflammatory mediators such as TNF-α, IL-1β, and IL-6, which contribute to cartilage degradation and pain sensitization while also modulating the senescence-associated secretory phenotype (SASP), a key driver of chronic joint inflammation [15,23,24].Matrix stabilization: HA injections stimulate endogenous HA synthesis by synoviocytes and enhance the integrity of the cartilage matrix by reducing enzymatic degradation, including the inhibition of matrix metalloproteinase-13 (MMP-13) which plays a major role in cartilage breakdown [8,11,25,26].

Recent studies suggest that HA may also exert senomorphic effects, meaning it can modulate cellular aging and slow down chondrocyte senescence, potentially delaying OA progression. While this aspect is not the primary focus of this review, it further underscores HA’s multifaceted role in joint homeostasis beyond its rheological impact.

HA is administered through intra-articular injection protocols, typically involving weekly injections over 3–5 weeks depending on the formulation. Some cross-linked HA formulations (hylan G-F 20) allow for single-dose administration [20,27,28,29,30].

#### HA Products in Clinical Use

HA products differ based on the molecular weight, cross-linking, and source of origin. The molecular weight of HA plays a critical role in determining its rheological and therapeutic properties, such as the following:Low molecular weight (0.5–1.0 × 10^6^ Da): shorter intra-articular retention but greater bioavailability (e.g., Suplasyn^®^, Fermathron^®^).Intermediate molecular weight (1.0–1.8 × 10^6^ Da): prolonged residence time, with balanced viscosity and anti-inflammatory effects (e.g., Ostenil^®^, Orthovisc^®^).High molecular weight (>6.0 × 10^6^ Da): increased viscoelastic properties and durability, though some studies suggest limited cellular bioavailability (e.g., Synvisc^®^, Synvisc-One™) [13,31].

HA can be avian-derived (e.g., rooster comb-based products) or bacterial-derived (fermented from *Streptococcus* species), with the latter having a lower risk of allergic reactions [30,32].

Despite its benefits, HA injections face significant challenges, particularly in the presence of joint infections, such as the following:Bacterial hyaluronidase degradation: pathogens such as *Staphylococcus aureus* produce hyaluronidases that degrade HA into smaller fragments, diminishing its protective effects [4,33].Inflammation-driven breakdown: oxidative stress and inflammatory cytokines accelerate HA depolymerization, reducing its retention in the joint space [34].Biofilm formation risk: degraded HA fragments can serve as bacterial adhesion sites, potentially promoting biofilm formation in septic joints [35].

These limitations have driven interest in polyacrylamide gels (PAAGs) as an alternative, as they demonstrate superior resistance to enzymatic degradation and prolonged intra-articular retention [31,36].

### 4.2. Polyacrylamide Hydrogels: A New Alternative?

Polyacrylamide hydrogels (PAAGs) have emerged as a promising alternative to traditional hyaluronic acid (HA)-based viscosupplements, particularly in cases where HA is rapidly degraded by enzymatic activity in the synovial environment [37]. Unlike HA, which has a limited residence time in the joint due to bacterial hyaluronidases and inflammatory degradation, PAAGs exhibit high stability, remaining in situ for extended periods while preserving their lubricating and shock-absorbing properties. Their cross-linked polymeric network allows for high water retention, mimicking the viscoelastic and tribological behavior of native synovial fluid [38]. Given these properties, PAAGs are increasingly being considered for patients with severe osteoarthritis and inflammatory joint diseases, where HA rapid breakdown limits its efficacy [39,40,41].

A key advantage of PAAGs is their ability to maintain stable rheological behavior under mechanical stress. While HA is highly shear-thinning—losing viscosity under load—PAAGs retain a more consistent gel-like structure, offering prolonged joint lubrication. This property is particularly relevant in high-load joints, where PAAGs effectively reduce friction and cartilage wear better than HA injections [10]. Additionally, PAAGs demonstrate superior tribological performance, lowering the coefficient of friction in artificial synovial environments and potentially extending the protective effects on cartilage.

Clinical trials have shown significant and prolonged symptom relief following PAAG injections. Unlike HA, which often requires multiple injections over short time intervals, PAAGs provide sustained therapeutic effects for up to 12 months, reducing treatment frequency and costs [38]. Their use is expanding beyond osteoarthritis, with growing interest in post-traumatic joint injuries and inflammatory conditions where synovial degradation is accelerated [42,43].

Despite these benefits, concerns remain regarding long-term biocompatibility and potential immune responses. Their prolonged residence time raises questions about whether persistent hydrogels could influence biofilm formation in infected joints. Unlike HA, which is gradually cleared from the synovial space, PAAGs persist, potentially creating a surface for bacterial adhesion in septic arthritis. Some studies have reported localized inflammation and swelling post-injection, necessitating further research into optimal formulations and administration protocols [10].

To mitigate these risks, next-generation PAAGs are being developed with antimicrobial properties or bioresorbable polymers to enhance safety in infection-prone joints [44]. Comparative studies between HA and PAAGs in both aseptic and septic conditions are crucial for establishing clear clinical guidelines. While PAAGs represent a major step forward in viscosupplementation, further research is needed to evaluate their long-term effects on joint health, immune interactions, and bacterial colonization dynamics. If these challenges are addressed, PAAGs could offer a durable, effective alternative to HA-based therapies for patients requiring sustained joint lubrication.

### 4.3. How Viscosupplementation Modulates Bacterial Proliferation and Biofilm Formation

The rheological properties of synovial fluid (SF) play a dual role in septic arthritis (SA) and periprosthetic joint infections (PJIs): they respond to bacterial colonization while simultaneously shaping the joint environment in ways that can either inhibit or promote microbial proliferation. This bidirectional relationship between SF rheology and bacterial metabolism is central to infection progression.

Pathogenic bacteria such as *Staphylococcus aureus* and *Staphylococcus epidermidis* exploit changes in SF by binding to plasma proteins, which act as adhesion sites for biofilm formation [8,13,45]. These biofilms mature into highly antimicrobial-resistant communities, persisting despite fluctuations in SF composition and often evading standard diagnostic methods, including synovial fluid aspiration and culture-based techniques [8,12].

The degradation of hyaluronic acid (HA), driven by bacterial hyaluronidases, reactive oxygen species (ROS), and host inflammatory mediators such as TNF-α and interleukins, significantly reduces SF viscosity [12,20]. As a result, the fluid becomes less effective in lubrication and immune defense, allowing bacteria to disperse more freely and settle onto joint surfaces where they can form biofilms [8,12]. However, bacterial adaptation to SF is not solely dependent on enzymatic degradation. The rheological properties of the fluid itself create an environment that can either enhance or inhibit microbial growth.

When SF becomes excessively diluted due to HA breakdown, the elastic modulus (G′) loses its plateau, the viscoelastic balance is disrupted, and bacterial motility increases. These changes favor bacterial dispersal within SF, promoting planktonic biofilm formation. The bacteria, now freely suspended in the low-viscosity SF, cluster into floating biofilm-like aggregates, exhibiting increased resistance to antibiotics and immune clearance, making their detection and eradication particularly challenging [8,46,47]. Studies indicate that such bacterial aggregation in SF can significantly influence the extent of colonization on prosthetic implants, further complicating treatment outcomes [48,49].Conversely, an excessive accumulation of HA, particularly following viscosupplementation, alters SF rheology in a manner that can also facilitate bacterial persistence [15,20]. When HA concentration increases beyond physiological levels (10 mg/mol or 5–10 mg/mL), SF transforms from a shear-thinning fluid into a viscoelastic solid-like material [48,49]. This shift is reflected in its rheological properties as the elastic modulus (G′) exceeds the viscous modulus (G″), meaning that the SF behaves more like a soft solid than a lubricating fluid [27,50]. High viscosity at low shear rates prevents effective fluid mobility, impairing joint lubrication rather than improving it. The polymeric network stabilizes, forming a dense meshwork that bacteria can exploit as an adhesion surface [51].

This solid-like transformation creates a quasi-surface for bacterial colonization, supporting the attachment of sessile biofilms rather than allowing bacterial dispersal [15,20]. Unlike planktonic biofilms, which float in low-viscosity SF, solid-like SF provides a stable environment for biofilm attachment and maturation.

In septic arthritis and post-surgical infections, this presents a serious clinical concern due to the following:○Bacteria can anchor themselves within the SF matrix, shielded from immune clearance and antimicrobial penetration [4].○Over time, the biofilm becomes highly resistant to treatment, increasing the risk of persistent infection and surgical failure [8,12].

Both low and high HA concentrations contribute to infection persistence through different biofilm behaviors, as summarized in Table 1.

These findings emphasize why standard microbiological cultures often fail to detect biofilm-embedded bacteria, leading to false-negative results and delayed treatment. Furthermore, they underscore the importance of maintaining an optimal synovial fluid balance, as both excessive HA depletion and over-supplementation can create environments that favor bacterial persistence and biofilm formation. The following will explore how advanced diagnostic techniques, including sonication and chemical pretreatment (DTT-based methods), can improve biofilm detection and guide more effective clinical management strategies.

## 5. Clinical Evidence and Open Questions

### 5.1. Challenges in Diagnosing and Treating Biofilm-Associated Infections

Diagnosing and managing biofilm-associated infections in septic arthritis (SA) and periprosthetic joint infections (PJIs) remains a significant clinical challenge. Biofilms are structured bacterial communities embedded in an extracellular polymeric substance (EPS) matrix, which enhances their adhesion to both implant surfaces and joint tissues while shielding them from host immune responses and antibiotics [52]. As biofilms mature, they intermittently disperse bacterial cells, promoting colonization of adjacent surfaces, which complicates eradication and increases the risk of persistent infections. A major clinical issue is the frequent failure of standard microbiological cultures to detect biofilm-associated bacteria, leading to false-negative results and delayed treatment [53]. Traditional diagnostic methods, such as synovial fluid aspiration, tissue sampling, and bacterial cultures, are often insufficient for detecting biofilm-embedded bacteria due to their sessile state and altered metabolic activity. Biofilm bacteria frequently produce small colony variants and dormant subpopulations that do not readily proliferate in standard laboratory conditions, further contributing to diagnostic failures [54].

This is particularly problematic in joint infections, where bacterial aggregates within synovial fluid behave similarly to surface-attached biofilms, exhibiting antibiotic resistance, protection from phagocytosis, and enhanced persistence [8]. While biofilms on implants have been extensively studied, fewer investigations have focused on synovial fluid biofilms, despite evidence that bacterial aggregates in SF share similar protective properties [55]. These clusters incorporate host components such as fibrin and hyaluronic acid, stabilizing bacterial communities and further complicating culture-based diagnostics [56].

To improve diagnostic accuracy, advanced techniques such as sonication and chemical pretreatment using dithiothreitol (DTT) have been proposed [57]. Sonication, a technique that employs high-frequency sound waves to dislodge bacteria from implant surfaces, has significantly improved microbiological recovery rates in prosthetic infections. However, its application is inherently limited to solid materials and is not feasible for processing synovial fluid, where bacterial aggregates remain suspended rather than adhered to a surface. A more promising approach for SF samples is dithiothreitol (DTT)-based chemical disaggregation, which targets disulfide bonds within the EPS matrix, effectively breaking apart bacterial aggregates and enhancing culture sensitivity [58]. This method has demonstrated improved diagnostic accuracy in implant-related infections (IRIs) and could be adapted for detecting biofilm-associated bacteria in synovial fluid [57] as it has already been extensively used in processing sputum samples from patients with respiratory diseases, functioning as a mucolytic agent that liquefies thick secretions to facilitate microbial identification [59,60,61]. Given its effectiveness in disrupting biofilm-like structures in complex biological fluids, its application in SF represents a rational and potentially valuable diagnostic tool. Despite these advancements, a standardized, highly sensitive method for biofilm detection in joint infections remains elusive. Further research is needed to optimize chemical treatment protocols for SF biofilm identification, particularly in high-viscosity environments where HA stabilizes bacterial aggregates.

Beyond localized joint infections, biofilm dispersion poses systemic risks. As bacterial clusters detach, they can enter circulation, increasing the likelihood of secondary infections such as infective endocarditis [62]. This risk underscores the clinical caution against orthopedic procedures in patients with active infections elsewhere, such as dental abscesses. The interplay between biofilm persistence, HA degradation, and chronic inflammation further impacts vascular health, potentially contributing to endothelial dysfunction. The following section explores these broader consequences and their implications for infection management.

### 5.2. Future Research Directions and Risk Mitigation Strategies

Emerging evidence underscores a strong interplay between biofilm-associated joint infections and systemic vascular impairment. Chronic inflammation in septic arthritis (SA) and periprosthetic joint infections (PJIs) leads to persistent immune activation, marked by elevated cytokines such as TNF-α, IL-1β, and IL-6, which also play a pivotal role in endothelial dysfunction and cardiovascular disease. These inflammatory mediators contribute to vascular impairment by promoting endothelial damage, increasing thrombogenicity, and accelerating atherosclerosis progression [18,19].

Additionally, biofilm-derived bacterial fragments entering the bloodstream raise concerns about distant complications such as infective endocarditis and septic embolism [63]. This is particularly relevant for patients with prosthetic heart valves or pre-existing cardiovascular conditions, as biofilm emboli from infected joint prostheses may exacerbate microvascular dysfunction, potentially leading to ischemic events [62,64].

Beyond infectious risks, alterations in synovial fluid (SF) composition may have systemic repercussions. The degradation of hyaluronic acid (HA) in SF, a hallmark of septic arthritis, could impair vascular homeostasis. The study by Mochizuki et al. (2003), highlights the role of HA glycosaminoglycans in shear-induced nitric oxide (NO) release from endothelial cells, a key mechanism in maintaining vascular tone [65]. The breakdown of HA may disrupt NO-mediated vasodilation, leading to endothelial dysfunction, reduced synovial perfusion, and impaired infection resolution.

Given these systemic implications, future research should prioritize the following:Developing biofilm-resistant viscosupplements: investigating HA derivatives or polyacrylamide-based hydrogels (PAAGs) that mitigate bacterial adhesion while preserving joint lubrication.Assessing vascular implications of SF alterations: exploring the impact of viscosupplementation on systemic inflammation, endothelial health, and cardiovascular risks.Advancing diagnostic strategies for vascular involvement: investigating imaging techniques and biomarkers to detect early vascular changes in patients with chronic joint infections.

By bridging rheological and vascular research, future strategies can better address biofilm persistence and systemic complications and can optimize treatment approaches for joint infections.

### 5.3. Future Perspectives and Patient Stratification

A crucial aspect of optimizing viscosupplementation is refining patient selection criteria, ensuring that pretreatment screening for inflammatory markers and subclinical infections becomes standard practice. Recent EUROVISCO 2024 consensus guidelines emphasize the need for individualized treatment strategies in knee OA patients to maximize benefits while mitigating infection risks [66].

A risk-stratified approach should be considered, particularly for high-risk patient populations such as:Individuals with a history of joint infections, who may be more prone to biofilm-related complications.Patients undergoing immunosuppressive therapy, including rheumatoid arthritis patients on biologics and oncology patients.Post-surgical cases, where residual inflammation and surgical implants may increase susceptibility to infection.

Given the complexity of SF rheology and its interaction with bacterial dynamics, viscosupplementation should not be administered indiscriminately. Instead, a precision-based approach should be implemented, integrating arthrocentesis-based screening to exclude biofilm-associated infections before HA administration. Additionally, novel biofilm-resistant viscosupplements or modified formulations should be considered for high-risk patients, ensuring that therapeutic benefits outweigh infection risks.

Future research should also explore HA’s senomorphic potential in infected joints, assessing its dual role in both chondroprotection and immune modulation. Understanding this balance may refine viscosupplementation strategies to optimize both therapeutic efficacy and infection risk mitigation.

## 6. Conclusions

This review underscores the delicate balance of synovial fluid (SF) homeostasis and its critical role in joint health. Both HA depletion and excessive supplementation influence rheology, bacterial colonization, and biofilm formation, highlighting that viscosupplementation is not a neutral intervention. While it remains a valuable therapeutic strategy, it carries infection-related risks that necessitate careful patient selection and risk assessment. In pathological conditions such as septic arthritis (SA) and periprosthetic joint infections (PJIs), the degradation of HA and SF rheological alterations create an environment that can facilitate biofilm formation, complicating infection management. Conversely, excessive HA supplementation may increase bacterial adhesion, contributing to chronic infections and prosthetic failure. These findings emphasize the urgent need for a more precise, risk-adapted approach in viscosupplementation.

With growing evidence linking SF alterations to infection susceptibility and systemic risks, viscosupplementation should not be considered a one-size-fits-all therapy. Instead, a personalized, evidence-based approach should be adopted, integrating synovial diagnostics, rheological profiling, and biofilm detection tools to optimize joint protection while minimizing unintended complications. By shifting toward risk-adapted treatment protocols and pretreatment infection screening, clinicians can ensure that viscosupplementation remains a safe and effective intervention, particularly for patients at higher risk of complications.

## Figures and Tables

**Figure 1 microorganisms-13-00700-f001:**
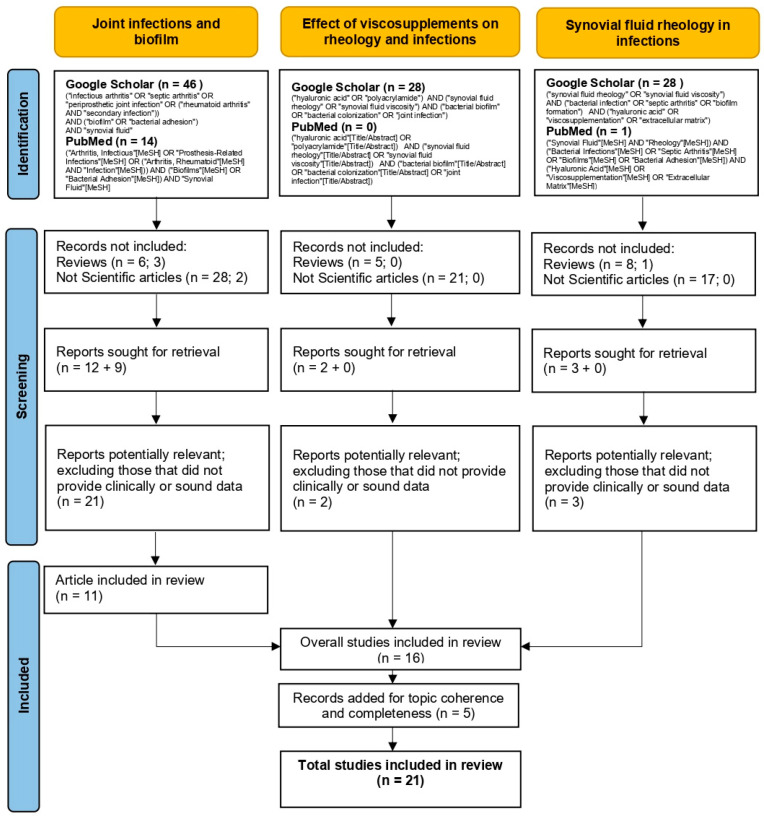
PRISMA flowchart illustrating the article identification, screening, and inclusion strategy used in this narrative review.

**Figure 2 microorganisms-13-00700-f002:**
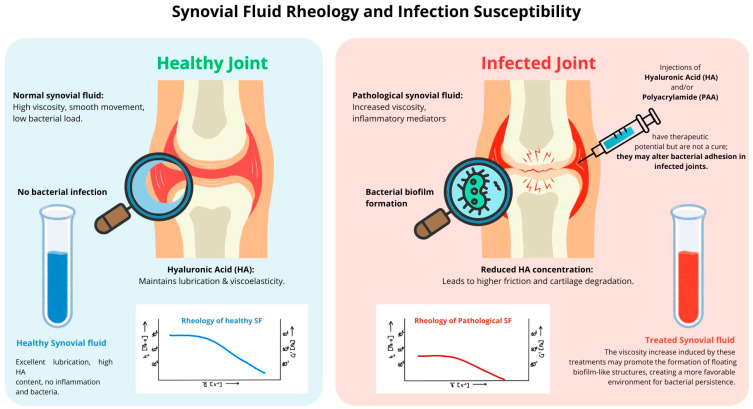
Schematic representation of synovial fluid alterations in healthy and infected joints, highlighting the impact of viscosupplementation on biofilm formation and rheological behavior. On the left, healthy synovial fluid (SF) ensures optimal lubrication, high hyaluronic acid (HA) content, and minimal bacterial presence. On the right, infected SF undergoes significant biochemical and rheological changes, including altered viscosity, increased inflammatory mediators, and bacterial biofilm formation. The introduction of HA and polyacrylamide (PAA) viscosupplements in infected joints may unintentionally modify bacterial adhesion and persistence, potentially promoting floating biofilm-like aggregates. Additionally, the figure includes representative rheological curves for healthy and pathological SF, providing a visual reference for the viscosity changes discussed in the text.

**Table 1 microorganisms-13-00700-t001:** This table summarizes the key rheological properties of synovial fluid (SF) in different pathological and treated conditions, highlighting their effects on bacterial proliferation and biofilm formation. This comparison underscores the importance of maintaining an optimal SF balance to prevent biofilm-associated infections.

Scheme	HA Concentration and MW	Rheological Behavior	Biofilm Fromation
Pathological synovial fluid (untreated)	Low HA concentration (<1–2 mg/mL), MW~$10^{6}\;g/mol$	Low viscosity, loss of shear-thinning, behaves as Newtonian fluid	Planktonic bacterial formation, increased bacterial motility and dispersion
Healthy synovial fluid	Normal HA concentration (2–4 mg/mL), MW~$10^{7}\;g/mol$	Non-Newtonian fluid, shear-thinning, viscoelastic balance	Minimal bacterial adhesion, homeostatic joint environment
Pathological synovial fluid (overtreated)	Excessive HA concentration (>5–10 mg/mL), MW remains high or increases	Solid-like behavior, reduced shear-thinning, increased elasticity	Potential biofilm adhesion due to high-viscosity and surface-like behavior

## Data Availability

No new data were created or analyzed in this study. This narrative review is based on previously published research, which has been cited appropriately throughout the manuscript. All data supporting the findings of this review are available in the referenced articles.

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
