# Peer review of "Viscosupplementation and Synovial Fluid Rheology: A Hidden Risk for Bacterial Biofilm Formation in Joint Infections?"

_microorganisms, 2025, doi:10.3390/microorganisms13040700_

Round 1
Reviewer 1 Report
Comments and Suggestions for Authors
This paper deals with the risk for bacterial biofilm formation and infections due to hyaluronic acid injections. It is interesting but many concerns have to be better addressed.
Firstly, both in the abstract and in the mani text there is no clarification about the review model you follow. Particularly, a method section should be provided to the readers. Which kind of review is it? In the case of a systematic review, you should integrate with the prospero registration code, you should assess a PRISMA flowchart, a PICO proposal, a quality assessment of the selected studies and a risk of bias evaluation. In the case of a narrative review, you should anyway provide some information about your search strategy. Which search string did you use? How did you select the scientific literature you reviewed? Which databases did you explore? A PRISMA flowchart is strongly recommended also in this case. Then, the organization of the search work should be described in the main text. In conclusion, a specifically dedicated method section should be inserted in the paper.
The presentation is good, but it seems that you underestimated the importance of hyaluronic acid as an effective treatment and a senomorphic agent for osteoarthritis. I suggest to briefly highlight these aspetcs according to the following scientific references:
-Farì, G., Mancini, R., Dell'Anna, L., Ricci, V., Della Tommasa, S., Bianchi, F. P., Ladisa, I., De Serio, C., Fiore, S., Donati, D., Ranieri, M., Bernetti, A., & Megna, M. (2024). Medial or Lateral, That Is the Question: A Retrospective Study to Compare Two Injection Techniques in the Treatment of Knee Osteoarthritis Pain with Hyaluronic Acid. Journal of clinical medicine, 13(4), 1141. https://doi.org/10.3390/jcm13041141
-Bernetti, A., Agostini, F., Paoloni, M., Raele, M. V., Farì, G., Megna, M., & Mangone, M. (2023). Could Hyaluronic Acid Be Considered as a Senomorphic Agent in Knee Osteoarthritis? A Systematic Review. Biomedicines, 11(10), 2858. https://doi.org/10.3390/biomedicines11102858
Best regards and good luck
Author Response
Dear Reviewer,
We sincerely appreciate your constructive feedback. Below, we provide detailed responses to each of your comments, indicating the corresponding modifications in the revised manuscript.
Comments 1: "This paper deals with the risk for bacterial biofilm formation and infections due to hyaluronic acid injections. It is interesting but many concerns have to be better addressed. Firstly, both in the abstract and in the main text there is no clarification about the review model you follow. Particularly, a method section should be provided to the readers. Which kind of review is it? In the case of a systematic review, you should integrate with the PROSPERO registration code, you should assess a PRISMA flowchart, a PICO proposal, a quality assessment of the selected studies, and a risk of bias evaluation. In the case of a narrative review, you should anyway provide some information about your search strategy. Which search string did you use? How did you select the scientific literature you reviewed? Which databases did you explore? A PRISMA flowchart is strongly recommended also in this case. Then, the organization of the search work should be described in the main text. In conclusion, a specifically dedicated method section should be inserted in the paper."
Response 1: Thank you for pointing this out. We agree with this comment. Therefore, we have explicitly stated in the manuscript that this is a narrative review, not a systematic review, and have included a dedicated "Materials and Methods" section to clarify our search strategy. This section now provides detailed information on:
- The databases explored (Google Scholar and PubMed).
- The specific search strings used.
- The criteria for article selection.
Additionally, while PRISMA flowcharts are primarily designed for systematic reviews, we acknowledge their utility in enhancing transparency. We have therefore adapted and included a PRISMA flow diagram to illustrate our search strategy, screening process, and inclusion criteria, despite the inherent challenges of applying it to a narrative review.
Furthermore, in response to the reviewers’ suggestions, we have expanded our dataset by including articles flagged as relevant. To reflect this, we have added a "Records added for topic coherence and completeness" box in the PRISMA flowchart (Figure 1). This ensures that the literature review is comprehensive and accounts for key references recommended during the review process.
These revisions can be found in:
- The newly added "Materials and Methods" section (page 2, paragraph 2).
- Figure 1, which presents the adapted PRISMA flowchart with the "Records added for topic coherence and completeness" box.
We sincerely appreciate your thoughtful feedback, which has helped improve the clarity and methodological rigor of our manuscript.
Comments 2: "The presentation is good, but it seems that you underestimated the importance of hyaluronic acid as an effective treatment and a senomorphic agent for osteoarthritis. I suggest to briefly highlight these aspects according to the following scientific references: https://doi.org/10.3390/jcm13041141 and https://doi.org/10.3390/biomedicines11102858 "
Response 2: Agree. We appreciate your suggestion and recognize the growing body of evidence supporting hyaluronic acid (HA) as a senomorphic agent in osteoarthritis. Accordingly, we have integrated references to HA’s senomorphic properties in the Introduction, as well as in Section 4.1 ("Hyaluronic Acid Injections: History, Mechanism, and Clinical Use"). However, given that the primary focus of this review is on synovial fluid rheology and its implications for bacterial adhesion and biofilm formation, we believe that dedicating an entire section to HA’s senomorphic role would be beyond the scope of this paper. Instead, we have ensured that this property is appropriately acknowledged in the context of HA’s broader biological effects, emphasizing its anti-inflammatory, chondroprotective, and potential anti-senescence roles. These revisions can be found in the Introduction (page 1, paragraph 1) and in Section 4 (page 6, paragraph 4.1), where HA’s senomorphic effects have been integrated into the discussion of its mechanism of action.

Reviewer 2 Report
Comments and Suggestions for Authors
The article addresses the important topic of how the rheological properties of synovial fluid affect the risk of bacterial biofilm formation in joint infections, with a particular focus on the viscosupplementation procedure. The paper is comprehensive, well-structured and includes a broad literature base. The authors attempt to provide a comprehensive analysis of the relationship between viscosupplementation therapy and the risk of developing joint infections, which is extremely important from the point of view of the diagnosis and treatment of joint diseases. The paper requires minor corrections and additions in terms of content and references before further prodecision. Please find my detailed notes and comments below.
Minor comments:
I feel that the introduction is far too short and needs to be supplemented with content on cartilage biomechanics and a more detailed description of joint fluid in the context of joint function. I recommend adding some references from recent years: Cartilage Integrity: A Review of Mechanical and Frictional Properties and Repair Approaches in Osteoarthritis; Short-term effects of arthroscopic microfracturation of knee chondral defects in osteoarthritis; Scale‐dependent rheology of synovial fluid lubricating macromolecules; Rheological behavior of the synovial fluid: a mathematical challenge; Investigation of the rheological and tribological characteristics of human synovial fluid;
The authors emphasize both the advantages and risks of viscosupplementation, but do not make clear recommendations for its use in cases of increased risk of infection. They should be clearer about when this treatment can be safe and when its use should be limited. Please clearly state your position on whether and when the use of viscosupplementation is safe, and in which cases it may pose an infectious risk.
Although the article mentions different types of viscosupplementation, it does not provide a detailed comparative analysis of their effects on synovial fluid rheology and infection risk. It would be useful to include comparative studies on HA degradation by bacterial enzymes under different conditions. I recommend a more thorough comparison of the properties of different types of viscosupplements in terms of their stability in an infected joint environment.
The authors point out the potential risks of viscosupplementation in the context of biofilms, but do not offer specific recommendations for strategies to minimize these risks. It would be useful to discuss, for example, potential antimicrobial additives for viscosupplementation or alternative application techniques. The article should include recommendations for methods to minimize the risk of infection, such as the use of viscosupplements with antimicrobial properties or optimization of administration techniques. This will significantly increase its scientific value.
The authors combine issues of biomechanics, microbiology and rheology, which makes the article valuable to a wide audience, including orthopedists, rheumatologists and specialists involved in the diagnosis of joint infections.
Author Response
Dear Reviewer,
We sincerely appreciate your constructive feedback and positive evaluation of our manuscript. We are particularly grateful for your recognition of the interdisciplinary nature of our work, which integrates biomechanics, microbiology, and rheology, making it valuable for orthopedists, rheumatologists, and specialists involved in joint infection diagnosis. Below, we provide detailed responses to each of your comments, indicating the corresponding modifications in the revised manuscript.
MAJOR COMMENTS
Comment 1: "The article addresses the important topic of how the rheological properties of synovial fluid affect the risk of bacterial biofilm formation in joint infections, with a particular focus on the viscosupplementation procedure. The paper is comprehensive, well-structured, and includes a broad literature base. The authors attempt to provide a comprehensive analysis of the relationship between viscosupplementation therapy and the risk of developing joint infections, which is extremely important from the point of view of the diagnosis and treatment of joint diseases. The paper requires minor corrections and additions in terms of content and references before further prodecision."
Response 1: Thank you for your positive evaluation of our work. We are pleased that you find the topic relevant, well-structured, and scientifically comprehensive. We have carefully addressed all your comments to enhance clarity, completeness, and the scientific value of our manuscript.
MINOR COMMENTS
Comment 2: "I feel that the introduction is far too short and needs to be supplemented with content on cartilage biomechanics and a more detailed description of joint fluid in the context of joint function. I recommend adding some references from recent years: Cartilage Integrity: A Review of Mechanical and Frictional Properties and Repair Approaches in Osteoarthritis; Short-term effects of arthroscopic microfracturation of knee chondral defects in osteoarthritis; Scale‐dependent rheology of synovial fluid lubricating macromolecules; Rheological behavior of the synovial fluid: a mathematical challenge; Investigation of the rheological and tribological characteristics of human synovial fluid."
Response 2: Thank you for pointing this out. We agree with this comment. Therefore, we have expanded the introduction by providing a more detailed description of cartilage biomechanics and synovial fluid function. We have also incorporated the suggested references to ensure a more comprehensive background discussion. These modifications can be found in Introduction (page 1 and 2, paragraph 1).
Comment 3: "The authors emphasize both the advantages and risks of viscosupplementation, but do not make clear recommendations for its use in cases of increased risk of infection. They should be clearer about when this treatment can be safe and when its use should be limited. Please clearly state your position on whether and when the use of viscosupplementation is safe, and in which cases it may pose an infectious risk."
Response 3: Agree. We have now explicitly clarified the conditions under which viscosupplementation is safe and when it should be avoided or used with caution. These changes can be found in the newly added Section 5 (page 11, paragraph 5.2.1).
Comment 4 and Comment 5: "Although the article mentions different types of viscosupplementation, it does not provide a detailed comparative analysis of their effects on synovial fluid rheology and infection risk. It would be useful to include comparative studies on HA degradation by bacterial enzymes under different conditions. I recommend a more thorough comparison of the properties of different types of viscosupplements in terms of their stability in an infected joint environment." and "The authors point out the potential risks of viscosupplementation in the context of biofilms, but do not offer specific recommendations for strategies to minimize these risks. It would be useful to discuss, for example, potential antimicrobial additives for viscosupplementation or alternative application techniques. The article should include recommendations for methods to minimize the risk of infection, such as the use of viscosupplements with antimicrobial properties or optimization of administration techniques. This will significantly increase its scientific value."
Response 4 and 5 : Thank you for your valuable feedback. We acknowledge the importance of understanding how different viscosupplements behave in an infected joint environment and how HA degradation by bacterial enzymes influences infection risk. However, as our review does not focus on a direct comparative analysis between different viscosupplements, we have chosen not to provide an extensive comparison of their stability profiles. Instead, we have addressed these concerns in Section 4 (page 8, paragraph 4.3) by discussing:
- The stability of HA in infected synovial fluid, particularly in relation to enzymatic degradation by bacterial hyaluronidases.
- The impact of HA degradation on bacterial adhesion and biofilm persistence.
- The rheological consequences of viscosupplementation in septic environments, particularly how excessive HA concentration may alter SF behavior, potentially creating conditions that favor bacterial colonization.
Additionally, we have emphasized the need for further research on viscosupplement formulations that minimize biofilm formation, including potential antimicrobial modifications and optimized administration techniques to reduce the risk of contamination. While we have not added a full comparative analysis of viscosupplements, we believe that Section 4.3 sufficiently addresses the concerns raised regarding stability, enzymatic degradation, and viscosupplementation in the context of biofilms.
Comment 6: "The authors combine issues of biomechanics, microbiology and rheology, which makes the article valuable to a wide audience, including orthopedists, rheumatologists and specialists involved in the diagnosis of joint infections."
Response 6: We sincerely appreciate your positive feedback on the interdisciplinary nature of our work. One of our key objectives was to bridge the gaps between biomechanics, microbiology, and rheology, making the findings clinically relevant to multiple specialties.
We have further refined the manuscript to ensure that the connections between these fields are clearly articulatedthroughout the text.

Round 2
Reviewer 1 Report
Comments and Suggestions for Authors
Dear authors,
thank you for the efforts to improve the quality of your paper according to my suggestions.
The article now is well structured and presented, each section is well organized and the conclusions are adequate.
It seems suitable for publication as it is, so no further corrections are needed.
Best regards